# Pitch Invariance Reveals Skill-Specific Coordination in Human Movement: A Screw-Theoretic Reanalysis of Golf Swing Dynamics

**DOI:** 10.3390/jfmk10030315

**Published:** 2025-08-15

**Authors:** Wangdo Kim

**Affiliations:** Ingenieria Mecanica, Universidad de Ingenieria y Tecnologia—UTEC, Lima 15063, Peru; mwdkim@utec.edu.pe

**Keywords:** pitch invariance, screw theory, skilled movement, biomechanical efficiency, motor coordination

## Abstract

**Background:** Skilled human movement, such as the golf swing, emerges from coordinated rotational and translational dynamics. This study investigates pitch—a screw-theoretic invariant defined as the ratio of linear to angular velocity along the instantaneous screw axis (ISA)—as a compact metric for quantifying motor coordination. **Methods:** We reanalyzed a validated motion capture dataset involving a proficient and a novice female golfer. ISA trajectories and pitch values were computed from 3D marker data, and synchronized with vertical ground reaction force (GRF) signals collected via force plate. **Results:** The proficient golfer exhibited tightly bounded pitch oscillations (approximately ±0.0025 cm/rad) that were temporally aligned with a single, well-defined GRF peak. In contrast, the novice showed irregular pitch fluctuations (−0.025 to +0.01 cm/rad) and asynchronous GRF patterns with multiple peaks. **Conclusions:** These findings demonstrate that pitch can serve as a biomechanical indicator of skilled performance, reflecting the degree of intersegmental coordination and force timing. Screw theory thus offers a rigorous framework for evaluating movement efficiency in sport and rehabilitation contexts.

## 1. Introduction

Human movement, particularly in skilled performance contexts such as the golf swing, arises from a rich interplay between translational and rotational motion [1,2]. The challenge of understanding such coordination lies in modeling not only individual joint actions but the overall structure and efficiency of the movement as a dynamic whole [3]. One of the most effective abstractions in this context is the rigid-body model—a simplification where connected anatomical segments are assumed to move cohesively, preserving distances between internal points [4]. Although biological systems are not perfectly rigid, the approximation holds remarkably well at the segmental level during high-speed ballistic actions such as a golf downswing [5].

Within this modeling framework, the motion of a rigid body is fully described by its instantaneous screw axis (ISA)—a spatial line along which translation and rotation are intrinsically coupled [6]. This axis is not fixed; it evolves as the movement progresses, especially in complex compound actions involving the whole body [3]. The theory of screws, originally formalized by Ball (1900), provides a unified geometric language for describing these motions through twists (velocity screws) and wrenches (force screws). This approach bypasses the limitations of traditional coordinate-based systems that rely heavily on angle derivatives and segmented force vectors [7].

In the golf swing, the body–club system exhibits a highly constrained rotational-translational motion, where the pitch—the ratio of linear to angular velocity along the ISA—emerges as a key biomechanical variable [6]. A constant pitch implies uniform screw motion and often correlates with skilled, efficient execution. Conversely, pitch irregularities reveal underlying asymmetries, disruptions, or compensatory patterns. Thus, pitch offers a powerful lens through which to evaluate motor coordination and segmental coupling, particularly in elite athletic contexts [3].

As illustrated in Figure 1, screw motion naturally gives rise to a helicoidal velocity field, where each point on the moving rigid body traces a path tangential to a helix aligned with the ISA [8]. The motion of any segment or limb can be decomposed into two components: a rotation about the ISA and a translation along it. This decomposition underpins not only the mathematical formalism of screw theory but also the embodied experience of coordinated motion—what skilled athletes often describe as smooth, connected, or rhythmic [9].

When applied to the downswing phase of a golf swing, this framework reveals how the club’s helicoidal path entrains and reflects pelvic rotation, not through explicit control, but through biomechanical necessity and inertial feedback [10]. By treating the club and the body as rigidly coupled, the analysis reduces to studying the evolution of pitch across time, grounded on the assumption that internal constraint forces (those preserving segment cohesion) do not perform work and may therefore be excluded from the virtual work equations (d’Alembert’s principle) [5].

In this study, we apply screw theory to reanalyze a previously recorded motion capture dataset of a professional and novice golfer. Using synchronized plots of pitch dynamics and vertical ground reaction force (GRF), we quantify the efficiency of segmental coordination during the downswing.

By focusing on the geometry of motion rather than the control signals themselves, our approach positions pitch as a perceptual–motor invariant [10]—a compact descriptor of how the body organizes itself relative to external affordances [11] like gravity, inertia, and ground contact. Through this lens, the rigid-body model is not a mechanical idealization but a perceptual-cognitive strategy for reducing control complexity in high-speed motor tasks. We conclude by exploring how this perspective may inform future applications in performance optimization, motor learning, and real-time biomechanical feedback systems.

## 2. Materials and Methods

### 2.1. Participants and Study Design

This study leveraged a previously validated biomechanical dataset [12] that recorded full-body golf swing dynamics of two female participants representing contrasting levels of expertise (Table 1).

The selection of this dataset was motivated by its comprehensive multi-modal instrumentation, which included high-speed motion capture synchronized with ground reaction force (GRF) data, providing a robust platform for advanced mechanical modeling.

The original data acquisition was conducted using a 12-camera Qualisys optoelectronic motion capture system (model: Oqus-300, Qualisys AB, Gothenburg, Sweden) operating at a sampling frequency of 300 Hz. A total of 24 retroreflective markers, in conjunction with 4 rigid-body clusters, were affixed to key anatomical landmarks following ISAK anthropometric protocols. This configuration enabled accurate 3D tracking of major body segments (e.g., head, thorax, pelvis, upper and lower limbs) and the golf club. Simultaneously, kinetic data were acquired via a Kistler force platform, aligned with the lead foot to define the global reference frame based on the initial center of pressure (COP).

Figure 2 illustrates the marker arrangement at the address phase, highlighting the detailed setup used for wrist and club tracking. These features were essential for subsequent analysis of grip torque dynamics and transmission of force impulses through the kinetic chain.

To focus the analysis on the most critical biomechanical phase of the swing, only the downswing portion—from the initiation of forward acceleration to just before club–ball contact—was examined. Using segment-based local coordinate systems, we computed the instantaneous screw axes (ISAs) of the club relative to the trunk, as well as the corresponding angular velocities and linear displacements.

The ISA trajectories [3] were interpreted as functional representations of the club’s inertial coordination with upper-body rotation. Using a screw-theoretic framework, the twist motion of the club was decomposed into time-varying pitch and orientation parameters, allowing us to characterize the dynamic coupling between translational and rotational components. This formulation enabled identification of control strategies used by each participant and provided a biomechanically grounded measure of skill-dependent swing organization.

This methodological choice to reuse an existing, high-quality dataset ensured internal consistency while reducing inter-session variability. It also preserved the ecological fidelity of the motor behavior, making it possible to examine how perception–action couplings unfold in real-world performance contexts. This approach is consistent with the theoretical framework adopted in our prior research on symmetry and motor optimization in skilled action.

### 2.2. Procedures and Data Acquisition

The spatial inertia tensor [12] $M\in R^{6\times 6}$ integrates both translational and rotational dynamics of a rigid body and is defined as(1)$$M=\left[\begin{array}{cc} I_{c}+m{\left[r_{c}\right]}^{\times }{\left[r_{c}\right]}^{\times } & m{\left[r_{c}\right]}^{\times }\\ -m{\left[r_{c}\right]}^{\times } & mI_{3} \end{array}\right]$$
where *m* is the mass, $r_{c}\in R^{3}$ is the position vector from the body frame to the center of mass, $I_{c}$ is the 3×3 rotational inertia tensor about the center of mass, ${\left[r_{c}\right]}^{\times }$ is the skew-symmetric matrix of $r_{c}$, and $I_{3}$ is the 3×3 identity matrix.

The associated eigenvalue problem [3] seeks screw vectors [7] $\theta \in R^{6}$, combining angular velocity ω and linear velocity v, that satisfy(2)$$M\;\theta =\mu \;\theta ,\;\theta =\left[\begin{array}{c} \omega \\ v \end{array}\right]$$

The pitch *h* of the screw is defined when ω≠0 as(3)$$h=\frac{v\cdot \omega }{\omega \cdot \omega }$$

This scalar represents the ratio of translation along the axis to rotation about it, encapsulating the geometric essence of the screw motion.

Furthermore, in the case of a finite twist, the pitch $h^{\#}$ can be derived from the following geometric components:(4)$$h^{\#}=\frac{LP^{\#}+MQ^{\#}+NR^{\#}}{L^{2}+M^{2}+N^{2}}=\frac{\Lambda \cdot \Pi^{\#}}{\Lambda \cdot \Lambda }$$
where $\Lambda ={\left[L,M,N\right]}^{⊤}$ defines the screw axis and $\Pi^{\#}={\left[P^{\#},Q^{\#},R^{\#}\right]}^{⊤}$ is the raw moment vector [7]. The component of $\Pi^{\#}$ along Λ isolates the pitch as follows:(5)$$\Pi =\Pi^{\#}-h^{\#}\Lambda$$

Although multiple coordinate choices for $\Pi^{\#}$ are possible, the screw axis remains uniquely defined. If $\Pi^{\#}$ is not colinear with Λ, the extracted pitch $h^{\#}$ alters the final moment vector Π, thereby changing the screw. Therefore, for the screw to represent a unique line in Plücker coordinates, $\Pi^{\#}$ must be proportional to Λ. This constraint ensures that the moment vector Π defines a consistent axis independent of parametric representation, as required in screw theory and spatial kinematics.

### 2.3. Signal Processing

According to Ball’s formulation of screw theory [6], any wrench acting on a rigid body constrained to twist about a given screw a can be represented as a component of a resultant wrench projected onto that screw. The principle of reciprocity states that for equilibrium to be maintained, the applied wrench must be reciprocal to the screw of allowable motion.

Formally, consider a system of wrenches acting on a body free to twist only about a given screw a. The total virtual work performed by the wrenches during an infinitesimal twist about a must be zero as follows:(6)$$P_{1}\left(\phi_{1}\delta \theta \right)+P_{2}\left(\phi_{2}\delta \theta \right)+\cdots +P_{n}\left(\phi_{n}\delta \theta \right)=0$$
which simplifies to(7)$$\sum_{i=1}^{n}P_{i}\phi_{i}=0$$

Here, $P_{i}$ represents the intensity of the *i*th wrench, and $\phi_{i}$ denotes the corresponding screw coordinate projection onto a. This condition ensures that the net effect of the applied wrenches does not disturb the constrained motion.

From this principle, Ball further notes that a given wrench can always be equivalently replaced by another wrench acting along a different screw, provided the body is constrained to twist only about a. This leads to the concept of a harmonic screw, whereby the applied and reactive wrenches achieve a dynamic balance across a screw system, allowing the body to undergo pure oscillation.

This harmonic relationship is foundational for modeling single-body oscillations—such as those seen in sports movements—where the effective action results from rhythmic alternation of forces applied on reciprocal screw systems [4]. The harmonic screw thus describes a self-consistent twist–wrench pairing that resonates with the mechanical affordances of the body and its constraints.

As Ball observed, this principle bears strong analogy to the classical condition for equilibrium of a particle constrained to a line under the action of multiple forces. If *P* and *Q* are two such forces acting at angles *l* and *m* to the direction of allowable motion, then(8)Pcosl+Qcosm=0

This analogy emphasizes that in both particle and screw systems, projected force components must cancel in the direction of allowable motion. The harmonic screw generalizes this idea to spatial rigid-body systems.

In the present study, we exploit this harmonic screw formulation to identify oscillatory modes in biomechanical motion—especially where the body pivots or twists rhythmically about a constrained axis. These modes provide insight into invariant biomechanical structures underlying efficient movement.

### 2.4. Computation of Instantaneous Screw Axis and Comparative Analysis

To identify the instantaneous screw axis (ISA) and spatial principal direction of inertia during the downswing, we used a screw-theoretic transformation framework based on rigid-body marker data. The motion segment (e.g., lower limb or club) was modeled as a rigid body defined by six non-collinear reflective markers tracked over time.

Let $X_{k}\left(t\right)$ and $X_{k}\left(t+\Delta t\right)$ denote the 3D coordinates of the *k*-th marker at two consecutive frames. The relative motion between time frames was computed using singular value decomposition (SVD) applied to centered marker configurations. Let A and B represent the positions at t+Δt and *t*, respectively. After removing the centroid offset, we computed the following:(9)$$H=\left(A-\bar{A}\right){\left(B_{0}-{\bar{B}}_{0}\right)}^{⊤}$$
where $\bar{A}$ and ${\bar{B}}_{0}$ denote mean marker positions. The rotation matrix R was extracted via the following:(10)$$H=UDV^{⊤},\;R=\left\{\begin{array}{cc} UV^{⊤}, & \mathrm{if}\;\det(UV^{⊤})>0\\ USV^{⊤}, & \mathrm{otherwise} \end{array}\right.$$
where S=diag(1,1,−1) ensures proper rotation. The translational shift T was derived from the mean displacement of marker centroids.

To incorporate spatial inertia, we mapped the original inertia tensor $I_{\mathrm{spatial}}$ into the moving frame via a screw transformation $S_{T}$ constructed as(11)$$S_{T}=\left[\begin{array}{cc} R & 0\\ {\left[T\right]}^{\times }R & R \end{array}\right],\;I_{\mathrm{transformed}}=S_{T}\;I_{\mathrm{spatial}}\;S_{T}^{⊤}$$

The eigensystem of the 3×3 rotational inertia submatrix $I_{rr}$ of the transformed inertia tensor was then computed to obtain the principal directions of rotation as follows:(12)$$I_{rr}v_{i}=\lambda_{i}v_{i}$$

To extract the eigen-screw, the cross-coupling term $I_{rv}$ was projected onto each eigenvector $v_{i}$, giving rise to the screw moment as follows:(13)$$\tau_{i}=I_{rv}v_{i},\;S_{i}=\left[\begin{array}{c} v_{i}\\ \tau_{i} \end{array}\right]$$

Among these, the direction corresponding to the minimum eigenvalue of the inverse inertia matrix was selected as the dominant principal screw, representing the minimal-energy configuration for rotational–translational coupling.

This computation was repeated for each frame following downswing initiation ($t=t_{0}=198$). The dominant screw axes were normalized and written to an output array, representing the time-evolving primary ISA during the downswing. These were exported as the matrix ‘Sprime.xlsx‘ for subsequent analysis and visualization.

The screw-based principal directions provide a biomechanically meaningful axis of movement coordination, invariant to translation and frame shifts, and consistent with screw theory’s geometric formalism.

## 3. Results

### 3.1. Instantaneous Screw Axis (ISA) Trajectories and Coordination Patterns

Figure 3 illustrates the trajectory of the instantaneous screw axes (ISAs) during the downswing phase for the proficient golfer. The ISA vectors form a tightly coiled helicoidal structure, indicating a well-regulated coupling between angular and linear motion. The consistency in both orientation and spatial placement of the screw axes suggests that the golfer’s body and club segment are rotating and translating in a stable, unified fashion. This pattern reflects efficient coordination across the kinetic chain, where internal joint torques are synchronized to produce smooth energy transmission along the instantaneous screw axis.

In contrast, Figure 4 shows the ISA trajectories for the novice golfer. Here, the screw axes are more dispersed, with visible discontinuities and abrupt directional changes. This irregularity reflects a breakdown in the coupling between rotational and translational motion, leading to inefficient momentum transfer. The variation in ISA alignment across frames indicates inconsistent segmental coordination and likely compensatory adjustments at the joint level, undermining the biomechanical efficiency of the swing.

### 3.2. Coupling Between Pitch Dynamics and Ground Reaction Force in Skilled and Novice Golf Swings

Figure 5 and Figure 6 illustrate the synchronized time series of pitch (top panels) and vertical ground reaction force (GRF, bottom panels) for the proficient and novice golfers, respectively. Pitch, expressed in cm/rad, was calculated as the ratio of translational to angular velocity along the instantaneous screw axis (ISA). GRF values were normalized to body weight and used to assess the timing and magnitude of lower-limb engagement during the downswing.

In the case of the proficient golfer (Figure 5), pitch values remain tightly bounded between −0.0025 and +0.0025 cm/rad, oscillating symmetrically around zero. This low-amplitude variation reflects a stable coupling between rotational and translational motion, consistent with a dynamically constrained rigid-body movement. The corresponding GRF trace reveals a single, smooth peak near 0.18× body weight, occurring at approximately 0.2 s. The temporal alignment between this GRF impulse and the most stable phase of the pitch profile indicates effective coordination across the kinetic chain. Lower-limb force application is precisely synchronized with upper-body angular–linear integration, facilitating efficient energy transfer toward impact.

By contrast, the novice golfer (Figure 6) exhibits a markedly different pattern. The pitch curve shows broad fluctuations ranging from −0.025 to +0.01 cm/rad, including frequent sign reversals and abrupt transitions. These variations signal unstable rotational–translational coupling, likely caused by inconsistent joint torque regulation and poor segmental coordination. The GRF profile further highlights this instability: multiple peaks and irregular timing reflect uncoordinated lower-limb force application. There is no clear alignment between the GRF impulse and any period of pitch stabilization, suggesting that the novice golfer fails to synchronize internal dynamics with external ground interaction.

Together, these results confirm that the degree of pitch stability and its temporal alignment with GRF patterns serve as robust indicators of motor coordination. The proficient golfer demonstrates a well-integrated perception–action system, whereas the novice displays biomechanical fragmentation and inefficiency.

### 3.3. Pitch Invariance and Perceptual-Motor Integration

Taken together, the data support the role of pitch as a perceptual–motor invariant—an emergent property of skilled coordination rather than a controlled parameter [13]. The stability of pitch in the skilled golfer reflects a self-organizing system [14] that minimizes unnecessary degrees of freedom by aligning with the geometric constraints of the motion. Conversely, the novice golfer’s variable pitch profile reveals instability in exploiting such constraints, resulting in inefficient compensatory strategies.

These results reinforce the hypothesis that pitch serves as a higher-order descriptor of movement quality, encoding both biomechanical coherence and temporal alignment with task-relevant external forces. As a scalar invariant derived from screw theory, pitch provides an elegant and computationally tractable metric for assessing movement skill in dynamic, whole-body actions such as the golf swing.

## 4. Discussion

This study applied screw-theoretic analysis to reexamine skill-level differences in golf swing dynamics, focusing on the behavior of pitch—the ratio of linear to angular velocity along the instantaneous screw axis (ISA). The key finding is that pitch functions as a robust biomechanical invariant, capable of capturing the temporal and spatial coherence of segmental coordination in a high-speed, multi-joint action.

### 4.1. Pitch as a Marker of Coordinated Efficiency

The skilled golfer consistently maintained pitch within a narrow, low-amplitude range, oscillating near zero. This pattern reflects a dynamic balance between rotational torque and linear translation, indicative of efficient energy transmission through the kinetic chain [15]. The observed stability in pitch suggests that the golfer’s body–club system behaves, in effect, as a unified rigid body during the downswing phase. This rigid-body-like behavior allows for minimal internal work, optimized momentum transfer, and predictable impact timing.

In contrast, the novice exhibited erratic pitch variation, characterized by frequent sign changes and large magnitude fluctuations. These findings reveal a failure to stabilize the coupling between angular and linear components of motion. Such instability may arise from inconsistent joint torque generation, mistimed segmental coordination, or an inability to exploit the body’s mechanical constraints effectively.

### 4.2. Harmonic Screw Alignment and Mechanical Resonance

From a mechanical standpoint, pitch directly encodes the helicoidal structure of rigid-body motion, with constant pitch indicating a uniform coupling of translation and rotation about a moving axis. In the skilled golfer, the narrow-band oscillations in pitch suggest a close approach to the *harmonic screw condition* described by Ball, in which the pitch of the twist equals the pitch of the wrench. Under this condition, the twist and wrench are aligned such that the virtual work is maximized without internal dissipation, allowing external impulses—such as the ground reaction force—to reinforce the motion rather than resist it. This resonance-like state yields a self-sustaining mechanical efficiency, reducing energy leakage and enhancing repeatability.

The novice’s high variability in pitch reflects repeated departures from this harmonic alignment, forcing compensatory joint actions and dissipating energy into non-productive degrees of freedom.

### 4.3. Pitch as a Perceptual Invariant

From an ecological perspective, pitch invariance operates as what Gibson termed a higher-order invariant: a stable relationship in the sensory flow that remains constant despite fluctuations in lower-level sensory variables. Here, the invariant emerges from the co-organization of body segments around the ISA, whose migration is shaped by both inertial and gravitational constraints. Skilled performers appear attuned to this invariant, allowing them to exploit environmental affordances—such as the ground’s reactive capability—without micromanaging each joint or segment. This aligns with the concept of *visual kinesthesis*, in which perception and action are tightly coupled through biomechanical invariants.

### 4.4. Applied Implications

The relationship between ISA geometry and pitch invariance opens practical possibilities for performance assessment and skill acquisition. Since pitch is a scalar quantity computable from motion capture or IMU data, it can serve as a compact and interpretable feedback metric. Training interventions could target stabilizing pitch during key phases, using haptic or visual cues to reinforce optimal angular–linear coupling. While demonstrated here in golf, the same principle may apply to other ballistic or high-speed actions, such as baseball pitching, tennis serving, or military tasks involving precise weapon handling, where efficient twist–wrench coupling is critical.

### 4.5. Broader Relevance

Finally, this work illustrates how screw-theoretic dynamics can bridge mechanical analysis and perceptual–motor theory, offering a unifying framework for understanding skilled performance. By integrating harmonic screw theory with Gibson’s ecological approach, pitch invariance emerges not only as a biomechanical signature of expertise but also as a perceptual anchor for coordinating action in complex environments.

## 5. Conclusions

This study applied screw-theoretic analysis to identify pitch—the ratio of linear to angular velocity along the instantaneous screw axis (ISA)—as a biomechanical invariant that reveals the quality of motor coordination in golf swing performance. The comparison between a proficient and a novice golfer showed that skilled performance is characterized by stable, low-amplitude pitch variation synchronized with well-timed ground reaction forces. In contrast, the novice demonstrated erratic pitch behavior and temporal misalignment, indicating disrupted segmental coordination and inefficient kinetic sequencing.

Achieving pitch stability is not merely a matter of local joint control but requires the golfer’s entire body to adapt continuously to the evolving spatial structure of the swing. Whether interacting with the shape of the environment, the contour of the swing path, or the forces being generated internally, the golfer must coordinate multiple joint functions in a fluid and integrated manner. Anatomically, pitch regulation demands dynamic blending across muscle groups, enabling the body to mold itself in response to inertial demands. In this way, the golfer becomes actively attuned to the form of space—not just reacting to it, but co-creating it as a functional environment for efficient movement.

These findings reinforce the value of pitch as a perceptual–motor invariant, linking biomechanical organization with task-specific adaptability. Future work may explore how training protocols or real-time feedback systems can help athletes learn to stabilize pitch through improved kinetic timing, intersegmental coupling, and ecological sensitivity to the dynamic constraints of the swing.

## Figures and Tables

**Figure 1 jfmk-10-00315-f001:**
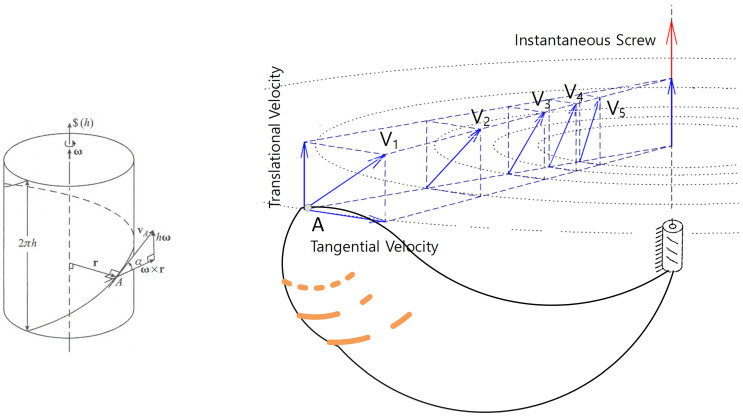
Helicoidal velocity field generated by screw motion. Each point traces a helical path aligned with the instantaneous screw axis (ISA), combining rotation about and translation along the axis.

**Figure 2 jfmk-10-00315-f002:**
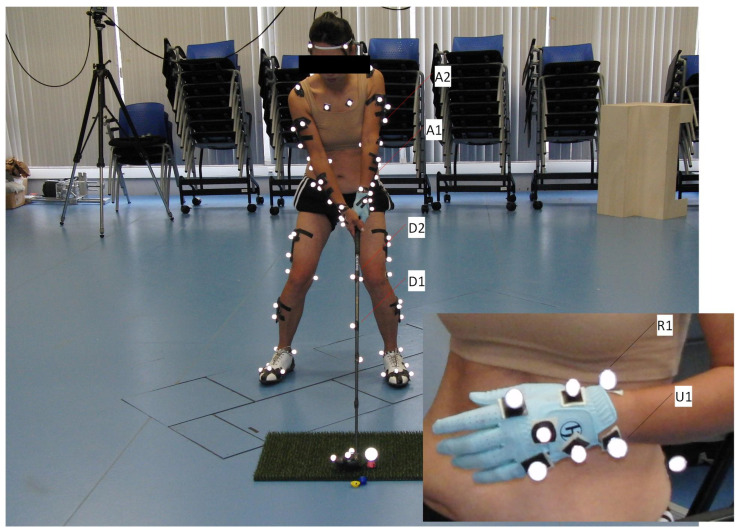
Reflective marker placement during address position. Detailed configuration shown for wrist and club tracking.

**Figure 3 jfmk-10-00315-f003:**
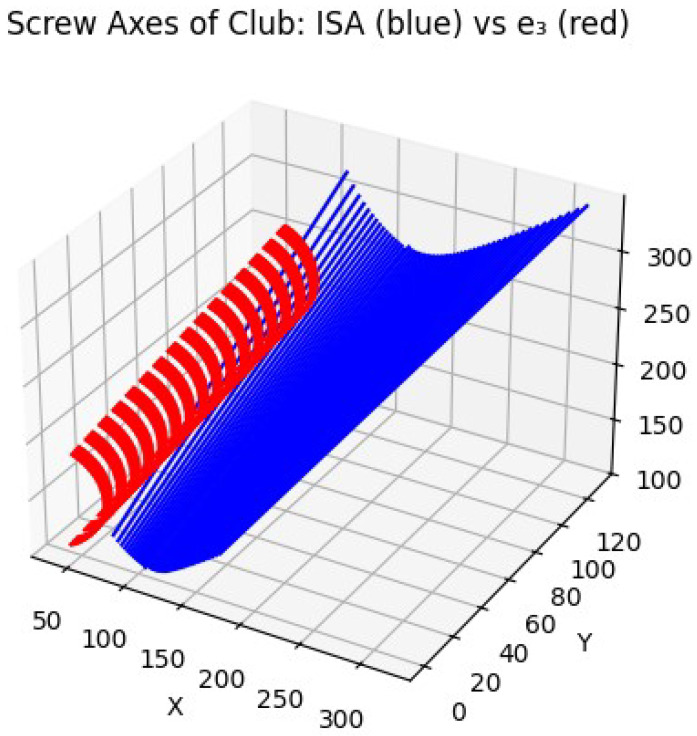
ISA trajectories for the proficient golfer. The helical structure of the screw axes indicates stable coupling of rotational and translational motion during the downswing. Here, $e_{3}$ denotes the principal axis of the club’s inertia tensor, representing its dominant rotational direction. Comparison with the ISA illustrates the degree of alignment between the club’s inertial properties and the actual screw axis of motion during the downswing.

**Figure 4 jfmk-10-00315-f004:**
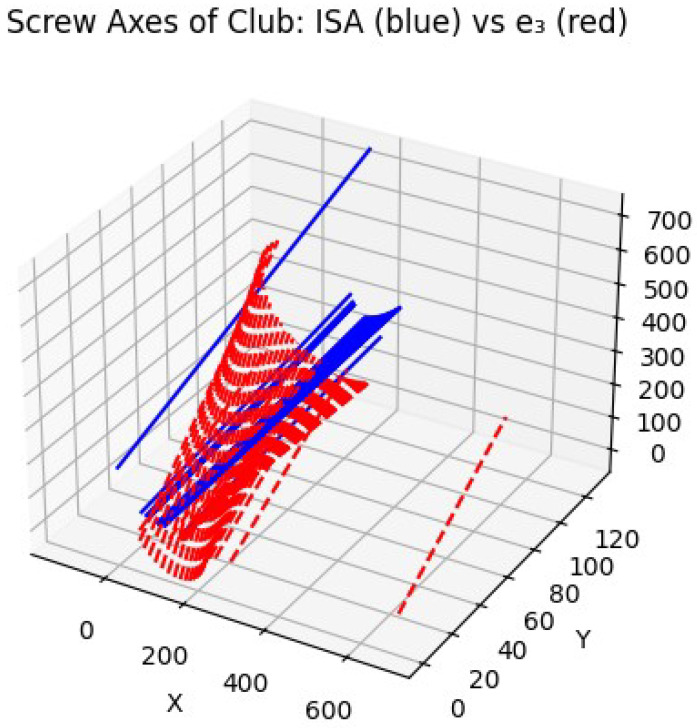
ISA trajectories for the novice golfer. Discontinuities and spatial disorganization of the screw axes reflect unstable motion coordination. Here, $e_{3}$ denotes the principal axis of the club’s inertia tensor, representing its dominant rotational direction. Comparison with the ISA illustrates the degree of alignment between the club’s inertial properties and the actual screw axis of motion during the downswing.

**Figure 5 jfmk-10-00315-f005:**
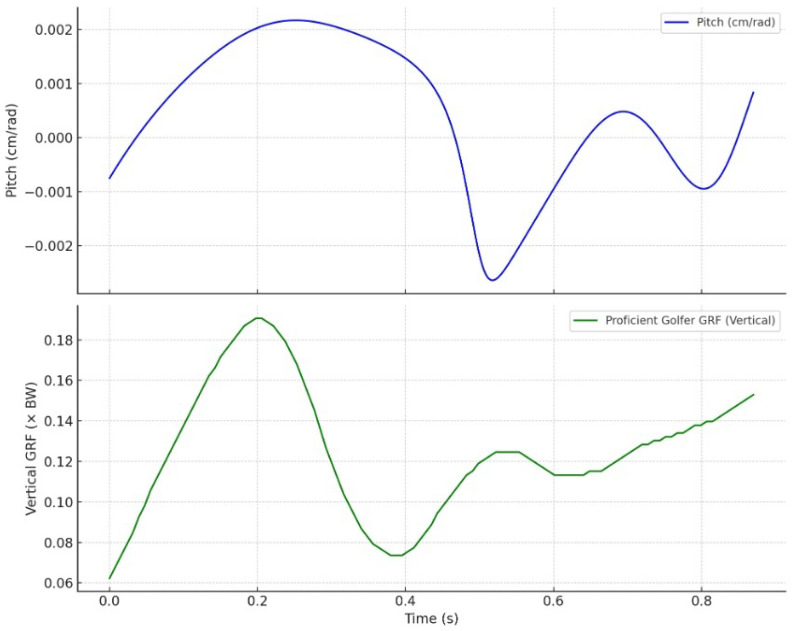
Proficient golfer: synchronized plots of pitch ((**top**), in cm/rad) and vertical ground reaction force ((**bottom**), normalized to body weight). A narrow, symmetric pitch profile aligns with a smooth GRF peak at approximately 0.2 s, indicating coordinated angular–linear coupling and effective lower-limb engagement.

**Figure 6 jfmk-10-00315-f006:**
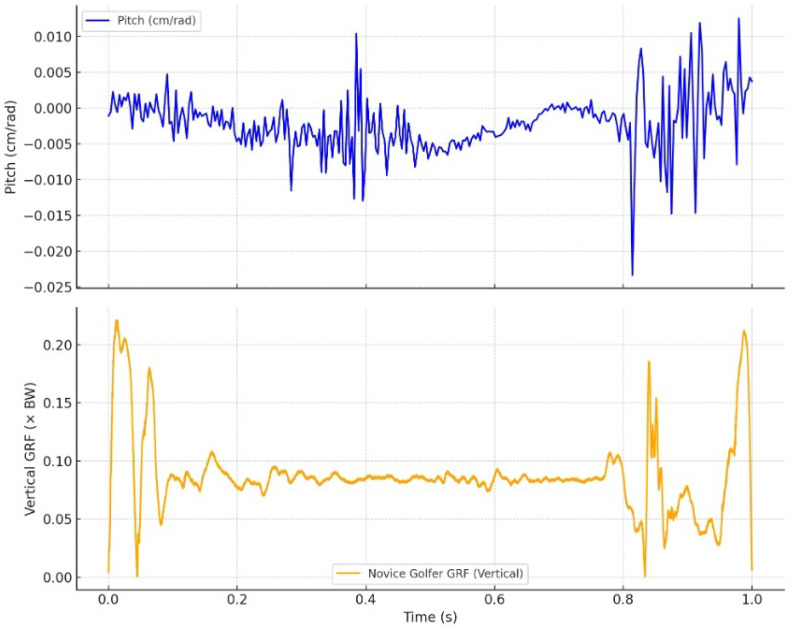
Novice golfer: pitch (**top**) and vertical GRF (**bottom**). The pitch profile exhibits large fluctuations and discontinuities, while the GRF curve shows multiple irregular peaks and poor timing alignment. This reflects unstable segmental coordination and reduced efficiency.

**Table 1 jfmk-10-00315-t001:** Participant demographics and golfing background. Key performance indicators include age, height, mass, years of experience, and golf handicap.

Participant	Age (Years)	Height (cm)	Mass (kg)	Handicap	Experience (Years)	Rounds/Year
Proficient Golfer (A)	17	167	54	32	1	10
Novice Golfer (B)	51	165	55	8	15	110

## Data Availability

The data presented in this study are available upon request from the corresponding author. The original contributions presented in this study are included in the article. Further inquiries can be directed to the corresponding author.

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
