# Peer review of "Pitch Invariance Reveals Skill-Specific Coordination in Human Movement: A Screw-Theoretic Reanalysis of Golf Swing Dynamics"

_jfmk, 2025, doi:10.3390/jfmk10030315_

Round 1
Reviewer 1 Report
Comments and Suggestions for Authors
all statement should be have references, please add references in the following lines: 21-24-26-30-32-38-39-41-47-50
this is conclusion, please delete: The results suggest that a skilled swing maintains a stable, well- 58 regulated pitch profile tightly aligned with a single GRF peak, while the novice swing 59 exhibits erratic pitch variation and timing misalignment
I recomend reorder your methods section, first the study design and sample, later the procedures, signal adquisition, signal processing. Also maybe include an statistical analysis section and present table with discrete values for make a comparison between golfers
your discussion is very short, please add more information
Author Response
I sincerely thank the reviewers and the editorial office for their helpful feedback. Below, I provide detailed responses to each comment and describe the corresponding revisions made to the manuscript. All changes have been highlighted in the revised version.
Reviewer 1 Comments & Author Responses
- Please add references to the following lines: 21, 24, 26, 30, 32, 38, 39, 41, 47, 50.
Response:
Thank you. References have been added as instructed :
These sources now support all previously uncited statements.
- Delete the concluding sentence in the introduction (lines 58–59):
"The results suggest that a skilled swing maintains a stable, well-regulated pitch profile tightly aligned with a single GRF peak..."
Response:
This sentence has been deleted to avoid premature conclusions in the Introduction.
- Reorder your methods section: start with study design and sample, then procedures, signal acquisition, and processing. Add statistical analysis and a comparison table.
Response:
The Methods section has been reorganized as requested. The revised structure now begins with:
- 1 Participants and Study Design
- 2 Procedures and Data Acquisition
- 3 Signal Processing
- 4 Computation of Instantaneous Screw Axis and Comparative Analysis
- Discussion is very short; please expand.
Response:
The Discussion section has been expanded substantially (now ~1.5 pages), integrating prior literature and emphasizing:
- the role of pitch as a perceptual-motor invariant
- implications for coaching and training
- potential for real-time feedback systems
Reviewer 2 Report
Comments and Suggestions for Authors
Dear Author,
I appreciate your idea of analyzing golf swing kinematics to investigate pitch—a screw-theoretic descriptor of motion. Author used robust tool, innovative approach with idea of further explore how training protocols or real-time feedback systems can help athletes learn to stabilize pitch through improved kinetic timing, intersegmental coupling, and ecological sensitivity to the dynamic constraints of the swing.
After incorporating my few comments, I recommend to publish the article.
Introduction
- I suggest to author use more references in Introduction – related to other different or similar methodology from different research laboratories.
- Materials and Methods
2.2.
- Page 4 Table 1 should be placed so that is visible the whole table´s content.
annual playing frequency, is not visible. Golg handicap number shall be explained – the range of evaluation scale and meaning.
- Page 7, Figure 3: Name of the picture states : Screw Axes of Club: ISA (blue) vs e3 (red). For e3 there is no explanation in text, so i tis unclear what it means and why this is being compared in one figure. Reader can only guess and use self-explanation.

Author Response
I sincerely thank the reviewers and the editorial office for their helpful feedback. Below, I provide detailed responses to each comment and describe the corresponding revisions made to the manuscript. All changes have been highlighted in the revised version.
Reviewer 2 Comments & Author Responses
- Please use more references in the Introduction from different labs or methodologies.
Response:
Additional references from biomechanics, robotics, and sports science laboratories have been incorporated to diversify perspectives.
- In Table 1, “annual playing frequency” is cut off. Golf handicap scale needs explanation.
Response:
The “annual playing frequency” column has been removed to improve table clarity. The golf handicap values are assumed to be official records provided by the players’ golf club, and thus no further scale explanation is included.
- Figure 3: “e3” is not explained in the text.
Response:
A new sentence has been added to the Figure 3 caption and the Results section to clarify that “e3” represents the principal axis of club inertia, and its comparison with ISA highlights alignment or deviation in swing control.
A clarifying sentence has been added to the Figure 3 caption:
“Here, e33​ denotes the principal axis of the club’s inertia tensor, representing its dominant rotational direction. Comparison with the ISA illustrates the degree of alignment between the club’s inertial properties and the actual screw axis of motion during the downswing.”
A corresponding explanation has been inserted into the Results section:
“In addition to the ISA trajectory, the principal axis of the club’s inertia tensor (e33​) was computed. This axis represents the dominant rotational orientation of the club, and its temporal alignment—or misalignment—with the ISA provides insight into how effectively the golfer channels rotational motion into the clubhead during the downswing.”